# Pulmonary and Renal Predictors of Mortality in ANCA-Associated Vasculitis: A Regional Experience from Türkiye

**DOI:** 10.3390/biomedicines13061401

**Published:** 2025-06-07

**Authors:** Dilara Bulut Gökten, Sevil Karabağ, Rıdvan Mercan

**Affiliations:** 1Department of Rheumatology, Tekirdag Namik Kemal University, Tekirdag 59059, Türkiye; 2Department of Pathology, Tekirdag Namik Kemal University, Tekirdag 59059, Türkiye

**Keywords:** ANCA-associated vasculitis, renal involvement, pulmonary involvement, mortality, predictor

## Abstract

**Background/Objectives**: Anti-neutrophil cytoplasmic antibody (ANCA)-associated vasculitis is a rare autoimmune disease marked by small-vessel inflammation. Pulmonary and renal manifestations are believed to critically influence prognosis, but detailed regional data are lacking. This study aimed to determine the prevalence and prognostic impact of pulmonary and renal involvement in AAV patients in the Thrace region of Türkiye. **Methods**: A retrospective cohort study was conducted on 78 biopsy-proven AAV patients followed between 2018 and 2025. Demographic, clinical, laboratory, and outcome data were analysed. Logistic regression identified predictors of relapse and mortality. **Results**: The cohort included 44 granulomatosis with polyangiitis, 30 microscopic polyangiitis, and 4 eosinophilic granulomatosis with polyangiitis patients; 40 were pr3-ANCA positive and 33 MPO-ANCA positive. Pulmonary involvement was observed in 71.8% and renal involvement in 74.4%, and overall mortality was 20.5%. All deaths occurred in patients with pulmonary involvement (28.6% vs. 0%, *p* = 0.048). Relapse was higher in those with pulmonary (17.9% vs. 4.5%, *p* = 0.048) and renal (15.5% vs. 5%, *p* = 0.056) involvement. Multivariate analysis showed that pulmonary involvement (OR 3.82, *p* = 0.002), renal involvement (OR 4.73, *p* = 0.013), and rituximab treatment (OR 10.79, *p* = 0.049) predicted relapse; elevated CRP (OR 1.01, *p* = 0.003), creatinine (OR 1.42, *p* = 0.028), hypoalbuminaemia (OR 0.24, *p* = 0.046), renal (OR 2.86, *p* = 0.031), and pulmonary (OR 3.21, *p* = 0.003) involvement predicted mortality. **Conclusions**: Pulmonary and renal involvement are highly prevalent and represent the strongest predictors of relapse and mortality in AAV patients in this regional cohort. Recognising these risks is essential to guide early interventions and improve patient outcomes.

## 1. Introduction

Systemic vasculitis refers to inflammation of the blood vessels and is classified based on the size of the affected vessels into small-, medium-, large-, and variable-vessel vasculitis [1]. Anti-neutrophil cytoplasmic antibody (ANCA)-associated vasculitis (AAV) is a rare group of diseases characterised by necrotising small-vessel inflammation and ANCA positivity, with a prevalence of fewer than 200 cases per million and an incidence below 25 per million person-years. This group includes granulomatosis with polyangiitis (GPA), microscopic polyangiitis (MPA), and eosinophilic granulomatosis with polyangiitis (EGPA) [2].

The pathogenesis involves neutrophil activation, the complement system, extracellular vesicles, and neutrophil extracellular traps (NETs) [3]. Typically, proteinase 3 (pr3)-ANCA and cytoplasmic (c-ANCA) patterns are found in GPA, whereas myeloperoxidase (MPO)-ANCA and perinuclear (p-ANCA) patterns dominate in MPA and about 40% of EGPA cases [4,5].

Clinically, AAV affects the upper and lower respiratory tracts, kidneys, eyes, and skin. It ranges from non-specific prodromal symptoms to severe, life-threatening organ dysfunction, such as diffuse alveolar haemorrhage or rapidly progressive glomerulonephritis [6]. Pulmonary involvement has been reported in 25–80% of cases, more commonly in GPA and EGPA (85–90%) than in MPA (25–60%) [7,8]. Radiological findings include interstitial pneumonia, nodules, cavitary lesions, and alveolar haemorrhage [9]. Renal involvement occurs in most MPA (90–100%) and 50–80% of GPA cases, with manifestations such as proteinuria, red blood cell casts, elevated creatinine, and reduced glomerular filtration rate (GFR) [10,11,12,13]

Both pulmonary and renal involvement are key prognostic markers. Even minimal renal impairment predicts poor outcomes, while patients needing renal replacement therapy have the worst survival [14]. Historically, untreated GPA had a one-year mortality rate near 80%, but advances in immunosuppressive therapy have improved survival, with current 10-year survival rates at ~80% for GPA and ~75% for MPA; EGPA has even better survival at 89–97% over five years [15,16].

Despite international data, regional evidence from Türkiye, especially the Thrace region, remains limited. This study aimed to evaluate demographic features, clinical presentations, laboratory findings, and predictors of poor outcomes in a biopsy-proven AAV cohort, focusing on the prognostic roles of pulmonary and renal involvement. By doing so, it sought to provide locally relevant insights to guide clinical practice.

## 2. Materials and Methods

This study was conducted as a retrospective cohort analysis. A total of 78 patients aged 18 years or older, who were followed in the rheumatology outpatient clinic of the study centre between January 2018 and December 2025 with a diagnosis of AAV, were included. The diagnosis of AAV vasculitis was based on the definitions proposed by the 2012 International Chapel Hill Consensus Conference on the Nomenclature of Systemic Vasculitis [17]. Patients were classified as having GPA, MPA, or EGPA according to the 2022 American College of Rheumatology (ACR)/European Alliance of Associations for Rheumatology (EULAR) classification criteria [18,19,20]. Patients were excluded from the study if they had insufficient clinical data, a follow-up duration of less than three months, or concurrent diagnoses of other systemic autoimmune diseases. The patients were classified into two groups based on pulmonary involvement as those with pulmonary involvement (PI) and those without (non-PI). Similarly, they were also categorised according to renal involvement into renal involvement (RI) and non-renal involvement (non-RI) groups. PI was defined based on thoracic CT findings consistent with vasculitis, such as interstitial pneumonia, pulmonary nodules, cavitary lesions, or alveolar haemorrhage. RI was defined based on renal biopsy-confirmed vasculitic involvement. Lung biopsies were performed only in select cases with uncertain radiographic or clinical features. ENT biopsies were performed in 40 cases. At diagnosis, the extent of disease activity was assessed using the third version of the Birmingham Vasculitis Activity Score (BVAS v3) [21]. ANCA testing was performed using the indirect immunofluorescence (IIF) method, and further subclassification into cytoplasmic ANCA (c-ANCA) and perinuclear ANCA (p-ANCA) was conducted through enzyme-linked immunosorbent assay (ELISA) [22,23]. Ethical approval for the study was obtained from the local ethics committee of the institution (approval number: 2025.60.03.18, approval date: 25 March 2025). As this was a retrospective analysis using anonymised patient data, the requirement for informed consent was waived by the committee. All data were handled in accordance with institutional policies to ensure the confidentiality and privacy of patient information.

Patient data were extracted from hospital records and included the following variables: AAV subtype, age at diagnosis, symptom duration prior to diagnosis, sex, and organ/system involvement at disease onset (including ENT, ocular, joint, cutaneous, renal, pulmonary, neurological, gastrointestinal, and cardiovascular systems). Additional information included comorbid conditions; ANCA subtypes and titres; and laboratory parameters at the time of diagnosis, such as erythrocyte sedimentation rate (ESR), C-reactive protein (CRP), serum creatinine, estimated GFR, alanine aminotransferase (ALT), aspartate aminotransferase (AST), albumin, haemoglobin, rheumatoid factor (RF), antinuclear antibody (ANA), and Birmingham Vasculitis Activity Score (BVAS). Laboratory thresholds were defined as follows: elevated CRP > 5 mg/L, elevated ESR > 20 mm/h, increased serum creatinine > 1.2 mg/dL, reduced estimated GFR < 60 mL/min/1.73 m^2^, and hypoalbuminaemia as serum albumin < 3.5 g/dL. Elevated AST and ALT levels were defined as >40 IU/L. The BVAS was calculated as a total composite score. Anaemia was defined as haemoglobin < 12 g/dL. RF and ANA positivity were determined according to institutional laboratory reference thresholds. Thoracic computed tomography (CT) findings were reviewed. Clinical follow-up data included records of relapse, remission, mortality, intensive care unit (ICU) admissions, dialysis requirement, and immunosuppressive therapies administered during the disease course. The study included patients who underwent organ biopsy, and renal biopsy findings were assessed using light microscopy and immunofluorescence techniques. Disease relapse was defined as the need for intensification of treatment in response to clinical and/or biochemical evidence of increased disease activity [24]. Remission was characterised by the complete resolution of both clinical signs and laboratory indicators of vasculitic activity [25]. Relapse and mortality were considered the primary outcome measures [26].

In this study, continuous variables were presented as either means ± standard deviations (SDs) or medians with interquartile ranges (IQRs), depending on the distribution of the data. The normality of continuous variables was assessed using the Kolmogorov–Smirnov and Shapiro–Wilk tests. For comparisons between two groups, the Student’s *t*-test was used for normally distributed data, whereas the Mann–Whitney U test was applied for non-normally distributed variables. When comparing more than two independent groups, one-way ANOVA was used under the assumption of normal distribution, and the Kruskal–Wallis test was preferred for non-normally distributed data. In cases where ANOVA showed a significant difference, Tukey’s Honestly Significant Difference (HSD) test was performed for post hoc pairwise comparisons. Categorical variables (such as sex, diagnosis category (GPA, MPA, and EGPA), ANCA subtype (pr3 and MPO), and mortality status) were expressed as frequencies and percentages. Group comparisons for categorical variables were carried out using the Pearson chi-square test, or Fisher’s exact test when expected cell counts were low. To identify factors associated with binary outcomes such as relapse and mortality, both univariate and multivariate logistic regression analyses were conducted. Initially, univariate logistic regression was used to examine the relationship between each independent variable and outcome. Variables that were statistically significant in univariate analysis were included in the multivariate logistic regression model to adjust for potential confounders. For each model, beta coefficients (βs), standard errors (SEs), Wald statistics, *p*-values, odds ratios (Ors), and 95% confidence intervals (Cis) were reported. All statistical analyses were performed using SPSS Statistics version 27.0 (IBM Corp., Armonk, NY, USA), and a *p*-value of <0.05 was considered statistically significant.

## 3. Results

### 3.1. Patient Demographics, Laboratory Findings, and Treatment Approaches

A total of 78 patients diagnosed with AAV were included in the study, of whom 44 had GPA, 30 had MPA, and 4 had EGPA. In total, 40 patients were positive for pr3-ANCA, 33 were positive for MPO-ANCA, and 5 were positive for both MPO-ANCA and pr3-ANCA. The median follow-up time was 38 months (IQR: 24–52 months), with a minimum of 6 months and a maximum of 84 months. The overall mortality rate in the cohort was 20.5%, with 16 patients dying during the follow-up period. Among the deceased patients, the majority (70.3%) died due to active vasculitis-related complications, such as pulmonary haemorrhage or renal failure, while the remainder succumbed to comorbid conditions, primarily cardiovascular disease or severe infections. The median age at diagnosis was 53.0 years (IQR: 26.0), and 48.7% of the patients were male (*n* = 38). The patients’ demographic characteristics, comorbidities, laboratory parameters at the time of diagnosis, and treatment modalities used during the follow-up are presented in Table 1 (see Table 1). Among the demographic characteristics, there was a statistically significant difference in the median age at diagnosis across the groups, with GPA patients being diagnosed at a younger age and MPA patients at an older age (*p* = 0.065). Post hoc analysis revealed that the significant difference in age was mainly between GPA and MPA patients (*p* = 0.012), while no significant differences were observed between GPA and EGPA or MPA and EGPA. A significant difference was also observed in sex distribution; male predominance was more common in GPA, while female patients were relatively more frequent in MPA (*p* = 0.448). The frequency of comorbidities showed a significant variation among the groups (*p* = 0.011), with hypertension being particularly more common in patients with MPA (*p* < 0.001). The higher hypertension rate in the MPA group was significantly different compared to the GPA group (*p* < 0.001) but not compared to EGPA. Regarding laboratory parameters, AST and ALT levels were significantly higher in the GPA group compared to the others (*p* = 0.035, *p* = 0.041). In terms of treatment approaches, cyclophosphamide was used more frequently in patients with MPA (*p* = 0.010).

### 3.2. Systemic Manifestations Across Vasculitis Subtypes

When systemic involvement patterns were evaluated, constitutional symptoms were common across all three vasculitis subtypes (69.2%), with no significant difference (*p* = 0.77). ENT involvement was significantly more frequent in GPA and EGPA compared to MPA (*p* < 0.000001), with sinusitis and chronic nasal discharge particularly prevalent in GPA and EGPA (*p* < 0.0001). ENT biopsies, performed in approximately 40 cases, confirmed granulomatous inflammation consistent with GPA. Pulmonary involvement was observed in 71.8% of patients, most commonly in GPA (90.9%) and EGPA (100%), showing a significant difference (*p* = 0.002). Cavitary lesions were exclusively observed in GPA (*p* = 0.002). All patients diagnosed with EGPA had a history of asthma and peripheral eosinophilia. Renal manifestations, including proteinuria and renal disease, were more prominent in MPA (*p* = 0.046, *p* = 0.05, respectively) (see Table 2).

When patients were divided into two groups based on the presence or absence of pulmonary involvement, 56 individuals (71.8%) were classified as having pulmonary involvement; those with pulmonary manifestations were more frequently male (*p* = 0.031), had significantly higher BVAS at presentation (*p* = 0.001), and were predominantly pr3-ANCA positive (*p* = 0.004) (see Figure 1). Conversely, MPO-ANCA positivity was more common in those without pulmonary involvement (*p* = 0.004). GPA was strongly associated with pulmonary involvement, whereas MPA predominated in the non-pulmonary group (*p* = 0.004). ENT involvement was also significantly higher in the pulmonary group (*p* = 0.014). All deaths occurred among patients with pulmonary involvement (28.6% vs. 0%, *p* = 0.048). The relapse rate was slightly higher in the pulmonary group (17.9% vs. 4.5%, *p* = 0.048). To illustrate the clinical diversity, one representative case from each of the groups of GPA, EGPA, and MPA patients with pulmonary involvement has been presented (see Figure 1).

When patients were stratified according to renal involvement, 74.4% exhibited kidney involvement (see Figure 2). Kidney involvement was defined based on renal biopsy findings. The majority of biopsies were consistent with crescentic glomerulonephritis (33.3%), while focal and sclerosing glomerulonephritis patterns were observed less frequently. These patients were significantly older at diagnosis (*p* = 0.023) and had lower haemoglobin levels (*p* = 0.033). MPO-ANCA positivity was more common in the renal group, while pr3-ANCA positivity predominated in those without renal involvement (*p* = 0.026 for both). ENT involvement was significantly more frequent in patients without renal involvement (90% vs. 37.9%, *p* = 0.008). Although not statistically significant, mortality tended to be higher in the renal group (51.7% vs. 5%, *p* = 0.0653), and relapse was more common in this group as well (15.5% vs. 5%, *p* = 0.056) (see Table 3). The median time to relapse was 17.6 months (range: 6.2–48.7 months), while the median time to death was 23.9 months (range: 8.1–60.5 months) from diagnosis. Among the 10 patients who experienced relapse, the majority presented with pulmonary manifestations (most commonly manifested as alveolar haemorrhage) (60.3%), followed by renal involvement (19.8%), ENT relapse (10.2%), and cutaneous vasculitis (9.7%). Illustrative renal biopsy samples are provided to demonstrate the histopathological spectrum (see Figure 2).

### 3.3. Predictors of Relapse and Mortality

Multivariate logistic regression identified pulmonary involvement (OR: 3.82; 95% CI: 2.06–22.61; *p* = 0.002), renal involvement (OR: 4.73; 95% CI: 1.40–15.94; *p* = 0.013), and rituximab treatment (OR: 10.79; 95% CI: 1.01–115.31; *p* = 0.049) as significant predictors of relapse. Factors associated with mortality included elevated CRP levels (OR: 1.01; *p* = 0.003), increased creatinine (OR: 1.42; *p* = 0.028), hypoalbuminaemia (OR: 0.24; *p* = 0.046), renal involvement (OR: 2.86; *p* = 0.031), and pulmonary involvement (OR: 3.21; *p* = 0.003) (see Table 4).

## 4. Discussion

This study provides valuable insights into the clinical spectrum and outcomes of AAV in a relatively large, well-characterised cohort. Pulmonary and renal involvement emerged as key determinants of both relapse and mortality, highlighting their critical role in disease prognosis. Notably, all observed deaths occurred in patients with pulmonary involvement, and those with renal manifestations exhibited higher relapse and mortality rates. A major strength of this study lies in its comprehensive clinical and laboratory evaluation, including biopsy-confirmed diagnoses and detailed organ-specific analyses.

In this cohort, GPA accounted for 56.4% of cases, followed by MPA (38.4%) and EGPA (5.1%). This distribution is consistent with previously published meta-analytic data, which identified GPA as the most frequent subtype globally, with pooled incidence rates of 9.0 per million person-years, compared to 5.9 for MPA and 1.7 for EGPA [27]. Compared to these large-scale studies, current findings reflect a similar dominance of GPA, though with relatively lower frequencies of EGPA, likely due to geographic, genetic, and environmental differences specific to the Thrace region of Türkiye.

When evaluated by AAV subtypes, MPO-ANCA was observed in 42.3% and pr3-ANCA in 51.2% of cases. This pattern differs from East Asian cohorts, where MPO-ANCA predominates, reflecting known geographic differences. In the GPA subgroup, pr3-ANCA positivity reached 90.9%, with ENT involvement in 81.8%, consistent with the classical GPA phenotype and prior reports linking pr3-ANCA with upper respiratory tract manifestations and distinct relapse patterns [28,29,30]. The median age at diagnosis was 53 years, with a noticeable age difference among subgroups: MPA patients had the highest median age (61 years), followed by EGPA (48.5 years) and GPA (44.5 years). Regarding treatment modalities, corticosteroids were administered to all patients in the present cohort. Cyclophosphamide was the most frequently used induction agent (66.7%), especially in MPA (93.3%), whereas rituximab was used in 30.8% of cases, more commonly in GPA (40.9%). Age and treatment patterns in this cohort were broadly consistent with prior Turkish and Colombian reports, although biologic use varied [31,32]. Despite differences in geography and healthcare settings, treatment strategies showed broad consistency.

Kidney involvement represents one of the most frequent and prognostically significant features of AAV. A markedly reduced GFR below 50 mL/min at presentation has been associated with an approximately 50% risk of progression to ESRD or death within five years [33]. In the present cohort, kidney involvement was observed in 74.4% of patients, characterised by higher MPO positivity, more frequent use of cyclophosphamide and plasmapheresis, and increased mortality and relapse rates, consistent with prior reports on renal-associated disease severity and outcomes [28,34]. Supporting this, pooled EUVAS trial data showed a 36.3% excess mortality at 20 years, with older age and reduced GFR identified as independent predictors of death. Findings from the present cohort are consistent with those of Binda et al., particularly regarding age, ANCA subtype, and organ involvement. Renal-involved patients were older at diagnosis (56 vs. 58.8 years), and MPO-ANCA was more frequent in this group, while pr3-ANCA and ENT involvement predominated among non-renal cases. These patterns support the association of pr3 positivity with extrarenal disease. Higher mortality and relapse rates in patients with kidney involvement, as seen in both cohorts, highlight the prognostic importance of renal dysfunction in AAV [35]. These findings collectively emphasise the strong association between kidney dysfunction and long-term outcomes in AAV [36].

Pulmonary involvement is a frequent and clinically significant feature of AAV, particularly in patients with GPA and pr3-ANCA positivity. In this study, pulmonary involvement was associated with higher mortality (28.6% vs. 0%) and relapse rates (17.9% vs. 4.5%) compared to those without pulmonary manifestations, aligning with the prior literature indicating that pulmonary complications increase the risk of poor outcomes. Sacoto et al. reported elevated mortality in AAV patients with lung involvement, with ICU mortality reaching up to 60.9% [37]. Additionally, the higher relapse frequency in this cohort may reflect the refractory nature of granulomatous pulmonary disease, often seen in pr3-ANCA-positive GPA. Although prior studies emphasised MPO-ANCA, this cohort showed a stronger link between pr3-ANCA and pulmonary manifestations [38,39]. Unexpectedly, despite prior reports highlighting the prognostic importance of ANCA subtype, pr3-ANCA positivity in this cohort showed only a partial association with adverse outcomes. This discrepancy may reflect regional differences in patient characteristics, referral patterns, or healthcare delivery in the Thrace region, underscoring the need to interpret findings within a local context. Furthermore, the strong link between pulmonary and renal involvement and mortality likely reflects their combined contribution to systemic inflammation, end-organ damage, and complications such as alveolar haemorrhage, respiratory failure, and irreversible renal injury. Elucidating the immunopathological mechanisms underlying these associations, including the roles of NETs, complement activation, and microvascular injury, warrants further investigation.

This study has several important limitations. First, its retrospective, single-centre design inherently carries the risk of selection and information biases, particularly given the centre’s status as a tertiary referral unit where more severe cases may be overrepresented. Second, the modest sample size, especially in the EGPA subgroup, limits the statistical power to detect subgroup-specific associations. Due to this limited sample size, only the most clinically and statistically relevant covariates were included in the multivariate models to minimise overfitting. Third, incomplete data, heterogeneity in diagnostic work-up (including biopsy practices), and limited standardisation in radiologic interpretation may have influenced the findings. Another limitation is the lack of detailed re-evaluation of pulmonary imaging, particularly regarding the presence of pulmonary nodules in MPA patients, which could reflect granulomatous disease typically seen in GPA. Finally, the absence of long-term follow-up data on chronic damage and quality of life restricts the assessment of broader patient outcomes. Future research should include larger, multicentre, prospective studies to validate these findings and explore regional and genetic influences on AAV presentation and prognosis. Additionally, mechanistic studies investigating the biological pathways linking pulmonary and renal involvement to mortality could help uncover novel therapeutic targets.

## 5. Conclusions

This study provides a comprehensive overview of the clinical characteristics, organ involvement patterns, treatment strategies, and outcomes of AAV in a regional cohort from the Thrace region of Türkiye. The findings underscore the critical prognostic role of pulmonary and renal involvement, both of which were significantly associated with higher relapse and mortality rates. pr3-ANCA positivity was closely linked to pulmonary and ENT involvement, particularly in GPA, while MPO-ANCA was more frequently associated with renal manifestations and MPA. Despite advances in immunosuppressive therapies, including the use of rituximab and cyclophosphamide, disease relapse and mortality remain important clinical challenges. Future prospective studies are needed to further clarify the impact of these organ involvements on long-term outcomes and to guide individualised treatment strategies.

## Figures and Tables

**Figure 1 biomedicines-13-01401-f001:**
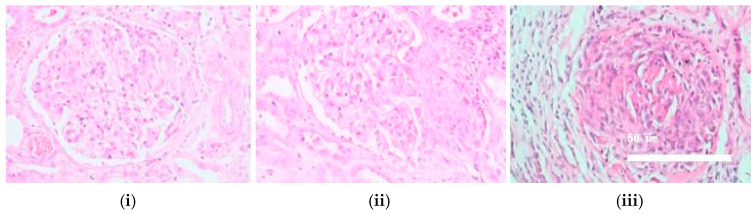
Histological images showing (**i**) a cellular segmental crescentic glomerulus from an EGPA case, (**ii**) a cellular crescentic glomerulus with glomerular necrosis from an MPA case, and (**iii**) a glomerulus with a cellular crescent and fibrinoid necrotising arteritis from a GPA case (H&E, ×400).

**Figure 2 biomedicines-13-01401-f002:**
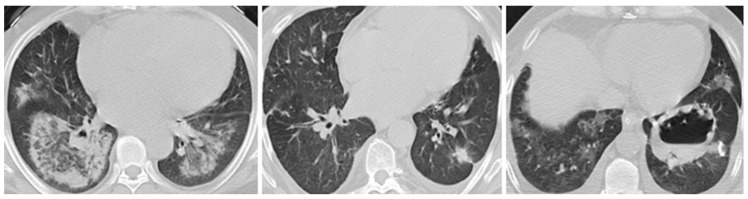
Chest computed tomography images from biopsy-proven cases showing pulmonary infiltrates in EGPA, pulmonary nodules in MPA, and cavitary lung lesions in GPA.

**Table 1 biomedicines-13-01401-t001:** Baseline demographic characteristics, laboratory findings, comorbidities, and immunosuppressive treatment approaches (including both induction and maintenance treatments) in patients with ANCA-associated vasculitis according to disease subtypes.

Variables	Total *n* = 78	GPA *n* = 44	MPA *n* = 30	EGPA *n* = 4	*p*-Value
Demographics					
Median age at diagnosis (years, IQR)	53.0 (26.0)	44.5 (19.5)	61.0 (9.5)	48.5 (16.5)	*0.006*
Sex (male), *n* (%)	38 (48.7)	28 (63.6)	8 (26.6)	2 (50)	*0.044*
Median duration of symptoms (years, IQR)	0.5 (0.7)	0.5 (0.7)	0.5 (0.8)	0.8 (0.2)	0.762
Smoking history, *n* (%)	26 (33.3)	18 (40.9)	8 (26.6)	0 (0)	1.00
Number of cases with comorbidity, *n* (%)	40 (51.2)	16 (36.3)	22 (73.3)	2 (50)	*0.011*
Comorbidities					
Diabetes mellitus, *n* (%)	16 (20.5)	4 (9.1)	12 (40)	0 (0)	0.100
Hypertension, *n* (%)	30 (38.4)	6 (13.6)	22 (73.3)	2 (50)	*<0.001*
Coronary artery disease, *n* (%)	6 (7.69)	2 (4.5)	4 (13.3)	0 (0)	0.719
Hyperlipidaemia, *n* (%)	8 (10.2)	2 (4.5)	6 (20)	0 (0)	0.445
Outcome					
ICU admission, *n* (%)	22 (28.2)	16 (36.3)	4 (13.3)	2 (50)	0.243
Dialysis, *n* (%)	40 (51.2)	20 (45.4)	18 (60)	2 (50)	0.685
Mortality, *n* (%)	16 (20.5)	12 (27.3)	2 (6.7)	2 (50)	0.178
Remission, *n* (%)	62 (79.5)	34 (77.3)	26 (86.7)	2 (50)	0.448
Relapse, *n* (%)	10 (12.8)	8 (18.2)	2 (6.7)	0 (0)	0.505
Laboratory values, mean ± SD, *n* (%)					
BVAS at presentation (mean ± SD)	19.67 ± 8.77	22.55 ± 10.02	16.13 ± 4.85	14.5 ± 7.78	0.063
ANA positivity, *n* (%)	18 (23.1)	8 (18.2)	8 (26.7)	2 (50)	0.543
RF positivity, *n* (%)	8 (10.3)	6 (13.6)	2 (6.7)	0 (0)	0.701
pr3 ANCA positivity, *n* (%)	40 (51.2)	40 (90.9)	0 (0)	0 (0)	1.000
MPO ANCA positivity, *n* (%)	33 (42.3)	0 (0)	30 (100)	3 (75)	1.000
Creatinine at presentation, mg/dL	3.14 ±2.63	2.79 ± 2.69	3.91 ± 2.56	1.17 ± 0.57	0.164
eGFR at presentation	48.49 ± 40.12	57.85 ± 44.92	32.29 ± 26.81	67.0 ± 46.67	0.291
AST (IU/mL)	20.53 ± 9.70	23.62 ± 10.43	16.87 ± 7.48	14.00 ± 2.83	*0.035*
ALT (IU/mL)	22.85 ± 12.45	26.41 ± 13.37	18.60 ± 10.28	15.50 ± 0.71	*0.041*
Albumin (g/dL)	3.65 ± 0.71	3.68 ± 0.70	3.68 ± 0.66	3.16 ± 1.48	0.799
Immune-suppressive treatment					
Corticosteroid, *n* (%)	78 (100.0)	88 (100.0)	60 (100.0)	4 (100.0)	1.000
Cyclophosphamide, *n* (%)	52 (66.7)	22 (50.0)	28 (93.3)	2 (50.0)	*0.010*
Azathioprine, *n* (%)	24 (30.8)	10 (22.7)	12 (40.0)	2 (50.0)	0.295
Methotrexate, *n* (%)	10 (12.8)	10 (22.7)	0 (0.0)	0 (0)	0.067
Mycophenolate mofetil, *n* (%)	27 (23.1)	10 (22.7)	8 (26.7)	0 (0)	1.000
Rituximab, *n* (%)	24 (30.8)	18 (40.9)	6 (20)	0 (0)	0.286
Plasma exchange, *n* (%)	14 (17.9)	12 (27.3)	2 (6.7)	0 (0)	0.204

GPA: granulomatosis with polyangiitis, MPA: microscopic polyangiitis, EGPA: eosinophilic granulomatosis with polyangiitis, IQR: interquartile range, BVAS: Birmingham Vasculitis Activity Score, ANA: antinuclear antibody, RF: rheumatoid factor, ANCA: antineutrophil cytoplasmic antibodies, pr3: proteinase 3, MPO: myeloperoxidase ICU: intensive care unit, eGFR: estimated glomerular filtration rate, ALT: alanine aminotransferase, AST: aspartate aminotransferase. *p*-values were obtained using chi-square or Fisher’s exact tests for categorical variables, and Kruskal–Wallis tests for continuous variables. *p* is italicized to indicate its statistical significance.

**Table 2 biomedicines-13-01401-t002:** Frequency of systemic manifestations in patients with ANCA-associated vasculitis according to disease subtypes.

	Total *n* = 78	GPA *n* = 44	MPA *n* = 30	EGPA *n* = 4	*p*-Value
Constitutional symptoms, *n* (%)	54 (69.2)	32 (72.7)	20 (66.7)	2 (50)	0.772
Neurologic involvement, *n* (%)	4 (5.1)	2 (4.5)	2 (6.7)	0 (0)	0.913
Arthralgia, *n* (%)	54 (69.2)	28 (63.6)	24 (80.0)	2 (50.0)	0.471
Arthritis, *n* (%)	10 (12.8)	16 (18.2)	4 (6.7)	0 (0)	0.630
Eye involvement, *n* (%)	10 (12.8)	10 (22.7)	0 (0)	0 (0)	0.114
Cutaneous, *n* (%)	6 (7.7)	4 (9.1)	2 (6.7)	0 (0)	0.881
ENT involvement, *n* (%)	40 (51.3)	36 (81.8)	0 (0)	4 (100.0)	*<0.000001*
Epistaxis, *n* (%)	8 (10.3)	8 (18.2)	0 (0)	0 (0)	0.132
Sinusitis, *n* (%)	32 (41)	28 (63.6)	0 (0)	4 (100)	*<0.0001*
Chronic nasal discharge, *n* (%)	30 (38.5)	28 (63.6)	0 (0)	2 (50)	*<0.0001*
Pulmonary involvement, *n* (%)	56 (71.8)	40 (90.9)	12 (40)	4 (100)	*0.002*
Alveolar haemorrhage/haemoptysis, *n* (%)	28 (35.9)	20 (45.5)	8 (26.7)	0 (0)	0.312
Respiratory failure, *n* (%)	14 (17.9)	12 (27.3)	2 (6.7)	0 (0)	0.202
Pleural effusion, *n* (%)	10 (12.8)	6 (13.6)	4 (13.3)	0 (0)	1.000
Pulmonary nodules, *n* (%)	40 (51.3)	26 (59.1)	12 (40)	2 (50)	0.321
Pulmonary infiltrates, *n* (%)	32 (41)	18 (40.9)	10 (33.3)	4 (100)	0.744
Cavitary lesions, *n* (%)	20 (25.6)	20 (45.5)	0 (0)	0 (0)	*0.002*
Renal impairment, *n* (%)	58 (74.4)	28 (63.6)	28 (93.3)	2 (50)	*0.046*
Haematuria, *n* (%)	54 (69.2)	26 (59.1)	26 (86.7)	2 (50)	0.143
Proteinuria, *n* (%)	58 (74.4)	26 (59.1)	30 (100)	2 (50)	*0.050*
Cardiovascular, *n* (%)	20 (25.6)	16 (36.4)	4 (13.3)	0 (0)	0.159
Pericardial effusion, *n* (%)	12 (15.4)	8 (18.2)	4 (13.3)	0 (0)	1.000

GPA: granulomatosis with polyangiitis, MPA: microscopic polyangiitis, EGPA: eosinophilic granulomatosis with polyangiitis, ENT: ear, nose, and throat. *p* is italicized to indicate its statistical significance.

**Table 3 biomedicines-13-01401-t003:** Comparison of clinical and laboratory characteristics according to pulmonary and renal involvement in patients with ANCA-associated vasculitis.

Variables	Total *n* = 78	PI *n* = 56	Non-PI *n* = 22	*p*-Value	RI *n* = 58	Non-RI *n* = 20	*p*-Value
Median age at diagnosis (years, IQR)	53.0 (26.0)	52.5 (27.5)	56.0 (18.0)	0.574	56.0 (18.0)	38.0 (22.8)	*0.023*
Sex (male), *n* (%)	38 (48.7)	34 (60.7)	4 (22.2)	*0.031*	24 (41.4)	14 (70)	0.155
Median duration of symptoms (years, IQR)	0.5 (0.7)	0.5 (0.7)	0.5 (0.5)	0.987	0.5 (0.75)	0.45 (0.2)	0.342
BVAS at presentation (mean ± SD)	19.7 ± 8.8	22.2 ± 9.0	13.1 ± 2.7	*0.001*	20.7 ± 9.1	16.6 ± 7.3	0.166
pr3 ANCA positivity	40 (51.2)	36 (64.28)	4 (18.18)	*0.004*	24 (44.4)	16 (88.9)	*0.026*
MPO ANCA positivity	32 (41.02)	14 (25)	18 (81.8)	*0.004*	30 (55.6)	2 (11.1)	*0.026*
GPA cases, *n* (%)	44 (56.4)	40 (71.4)	4 (18.1)	*0.004*	28 (48.3)	16 (80)	0.140
EGPA cases, *n* (%)	4 (5.1)	4 (7.1)	0 (0)	1.00	2 (3.4)	2 (10)	1.000
MPA cases, *n* (%)	30 (38.4)	12 (21.4)	18 (81.8)	1.00	28 (48.3)	2 (10)	1.000
ENT involvement, *n* (%)	40 (51.3)	36 (64.3)	4 (18.2)	*0.014*	22 (37.9)	18 (90)	*0.008*
Mortality, *n* (%)	16 (20.5)	16 (28.6)	0 (0)	*0.048*	30 (51.7)	2 (5)	*0.065*
ICU, *n* (%)	22(28.2)	20(35.7)	2(9.1)	0.130	18(31)	4(20)	0.693
Remission, *n* (%)	62 (79.5)	42 (75.0)	20 (90.9)	0.400	44 (75.9)	18 (90)	0.653
Relapse, *n* (%)	10(12.8)	10(17.9)	1 (4.5)	*0.048*	9(15.5)	1 (5)	*0.056*
AST (IU/mL), (mean ± SD)	20.5 ± 9.7	21.0 ± 10.4	19.4 ± 7.8	0.601	20.1 ± 10.1	21.9 ± 8.8	0.590
ESR (mm/h), (mean ± SD)	68.3 ± 31.1	68.9 ± 32.8	66.8 ± 27.9	0.844	74.0 ± 28.5	51.9 ± 33.9	0.087
CRP (mg/dL), (mean ± SD)	86.6 ± 72.9	90.4 ± 73.6	77.0 ± 73.7	0.616	88.2 ± 71.5	81.8 ± 80.8	0.827
Hemoglobin (g/dl), (mean ± SD)	11.1 ± 2.2	11.1 ± 2.3	10.9 ± 2.1	0.789	10.6 ± 2.0	12.4 ± 2.2	*0.033*

PI: pulmonary involvement, RI: renal involvement, IQR: interquartile range, BVAS: Birmingham Vasculitis Activity Score, pr3: proteinase 3, MPO: myeloperoxidase, GPA: granulomatosis with polyangiitis, MPA: microscopic polyangiitis, EGPA: eosinophilic granulomatosis with polyangiitis, ENT: ear, nose, and throat, ICU: intensive care unit, AST: aspartate aminotransferase, ESR: erythrocyte sedimentation rate, CRP: C-reactive protein, SD: standard deviation. *p* is italicized to indicate its statistical significance.

**Table 4 biomedicines-13-01401-t004:** Multivariate logistic regression analysis of factors associated with relapse and mortality in patients with ANCA-associated vasculitis.

Factors Related to Relapse	B	SE	Wald	*p*	Odds Ratio 95% CI
ESR	−0.226	0.192	−1.175	0.239	0.79 (0.547–1.163)
CRP	0.072	0.053	−1.351	0.176	0.93 (0.837–1.033)
Creatinine	2.166	1.964	1.102	0.270	8.73 (0.186–410.78)
Albumin	−5.236	3.753	−1.395	0.162	0.005 (0–8.33)
Cyclophosphamide	0.263	1.196	0.220	0.825	1.30 (0.124–13.57)
Rituximab	2.378	1.208	1.968	*0.049*	10.79 (1.010–115.31)
Renal	1.553	0.624	2.49	*0.013*	4.73 (1.40–15.94)
Pulmonary	1.918	0.625	3.07	*0.002*	3.82 (2.06–22.61)
ANCA titre	−0.008	0.012	−0.686	0.492	0.991 (0.967–1.01)
Age	0.061	0.091	−0.669	0.502	0.940 (0.78–1.12)
Sex	2.636	2.664	0.989	0.322	13.96 (0.07–2588.3)
**Factors Related to Death**	**B**	**SE**	**Wald**	** *p* **	**Odds Ratio** **95% CI**
ESR	0.010	0.013	0.775	0.438	0.010 (0.98–1.03)
CRP	0.009	0.007	1.40	*0.003*	1.01 (1.00–1.023)
Creatinine	0.33	0.19	1.75	*0.028*	1.418 (1.03–1.93)
Albumin	−0.64	0.81	−0.79	*0.046*	0.237 (0.05–0.97)
Cyclophosphamide	0.500	0.898	0.557	0.577	1.650 (0.28–9.60)
Rituximab	−1.348	1.133	−1.189	0.234	0.259 (0.02–2.39)
Renal	1.052	0.487	2.16	*0.031*	2.863 (1.10–7.40)
Pulmonary	2.220	0.743	2.99	*0.003*	3.21 (2.23–38.10)
Age	0.061	0.091	−0.669	0.402	0.540 (0.60–2.42)
Sex	−0.704	0.814	−0.865	0.386	0.494 (0.10–2.43)

ESR: erythrocyte sedimentation rate, CRP: C-reactive protein, B: beta coefficient (β), SE: standard error, CI: confidence interval. *p* is italicized to indicate its statistical significance. Bold text indicates the start of the death-related regression model.

## Data Availability

Data will be made available upon reasonable request.

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
