# Peer review of "Pulmonary and Renal Predictors of Mortality in ANCA-Associated Vasculitis: A Regional Experience from Türkiye"

_biomedicines, 2025, doi:10.3390/biomedicines13061401_

Round 1
Reviewer 1 Report
Comments and Suggestions for Authors
The authors present a retrospective analysis examining the association between pulmonary and renal involvement and mortality in ANCA-associated vasculitis (AAV) based on data from a regional medical center in Türkiye.
The topic is of clinical importance given the high morbidity and mortality associated with AAV, and the manuscript attempts to provide localized data that may complement global trends. The authors’ intent to emphasize organ-specific predictors of mortality is commendable and could be a valuable addition to the literature, especially from underrepresented geographic regions.
However, I regret to note that the manuscript in its current form requires substantial revision before it can be considered for publication. Several sections lack clarity and rigor, and critical methodological details are omitted. Additionally, the presentation of results and the depth of discussion fall short of the standards.
-
The abstract lacks specific data points, including quantitative mortality rates or statistical outcomes.
-
The objective and conclusion sections are somewhat vague and not tightly linked.
The introduction part, Introduces AAV and the relevance of pulmonary and renal involvement, but there is Lacks a focused narrative flow, with redundancy in disease descriptions and The novelty of the study is not clearly delineated — what does this study offer that others have not?
In material and methods part: The retrospective nature and inclusion/exclusion criteria are described but The time frame of patient data collection is not clearly specified and Statistical analysis is insufficiently described (e.g., was multivariate analysis performed?
The discussion lacks depth in interpreting unexpected findings and Limitations are acknowledged but understated — single-center, retrospective design, limited sample size, and possible selection bias are all significant. my Recommendations are Critically evaluate the implications of regional differences in care and outcomes and Expand on potential mechanisms linking renal/pulmonary involvement with mortality and then Strengthen the limitations section and suggest directions for future research.
The manuscript would benefit significantly from professional English language editing and Several grammatical errors, awkward phrasings, and inconsistent use of medical terminology are present.
Author Response
Reviewer 1: Thank you for your precious time and effort for reading my article. Your effort is really appreciated.
Comments 1: The abstract lacks specific data points, including quantitative mortality rates or statistical outcomes.
Response 1: The abstract was lacking specific data points, including quantitative mortality rates or statistical outcomes. We added quantitative mortality rates and statistical outcomes.
Comments 2: The objective and conclusion sections are somewhat vague and not tightly linked.
Response 2: The objective and conclusion sections were somewhat vague and not tightly linked, we corrected to maket hem more tightly linked. We highlighted that parts with yellow.
Comments 3: The introduction part, Introduces AAV and the relevance of pulmonary and renal involvement, but there is Lacks a focused narrative flow, with redundancy in disease descriptions and The novelty of the study is not clearly delineated — what does this study offer that others have not?
Response 3: The introduction part, lacked a focused narrative flow, with redundancy in disease descriptions and the novelty of the study was not clearly delineated, we added the novelty of this study, and deleted the redundant parts.
Comments 4: In material and methods part: The retrospective nature and inclusion/exclusion criteria are described but The time frame of patient data collection is not clearly specified and Statistical analysis is insufficiently described (e.g., was multivariate analysis performed?
Response 4: In material and methods part: The time frame of patient data collection was not clearly specified and Statistical analysis was insufficiently described, we added multivariate logistic regression analysis part and we specified the time frame of patient data collection.
Comments 5: The discussion lacks depth in interpreting unexpected findings and Limitations are acknowledged but understated — single-center, retrospective design, limited sample size, and possible selection bias are all significant. my Recommendations are Critically evaluate the implications of regional differences in care and outcomes and Expand on potential mechanisms linking renal/pulmonary involvement with mortality and then Strengthen the limitations section and suggest directions for future research.
Response 5: The discussion lacked depth in interpreting unexpected findings and limitations were acknowledged but understated. We evaluated the implications of regional differences in care and outcomes and expanded on potential mechanisms linking renal/pulmonary involvement with mortality and then strengthened the limitations section and suggested directions for future research.
Comments 6: The manuscript would benefit significantly from professional English language editing and Several grammatical errors, awkward phrasings, and inconsistent use of medical terminology are present.
Response 6: We have taken professional English language editing help and we corrected several grammatical errors, awkward phrasings, and inconsistent use of medical terminology.
Reviewer 2 Report
Comments and Suggestions for Authors
Manuscript with interesting results, but I have the following comments:
Please do not include abbreviations in the abstract, or mention what they mean: Results: The cohort included 44 GPA, 30 MPA, and 4 EGPA patients.
In summary, the following is unclear; please improve the following sentence: All deaths occurred in patients with pulmonary involvement (p = 0.048), and mortality tended to be higher in those with renal involvement (p = 0.065).
In the methods section, define what was called pulmonary involvement (PI) or renal involvement (RI).
Please mention if there was committee approval, its approval number and date, as well as whether informed consent was required or waived, and whether the confidentiality of personal data was guaranteed.
The description of the statistical analysis is very well done. Table 1. P values ​​of 1, 3, or 4 decimal places exist; please standardize them, also in the text. Please indicate which statistical test was used to obtain this P value.
Post-hoc analyses are not shown; please mention them.
As this is a cohort study, please define the minimum, maximum, and average follow-up time of the patients. Also, in cohort studies, the RR is shown, and you mention OR; please verify.
Please define everything you describe or analyze, for example, what you call elevated CRP levels, increased creatinine, hypoalbuminemia, etc.
3.4. Figures: It is not clear what you want to demonstrate, or the objective of this section; please expand.
Author Response
Thank you for your precious time and effort for reading my article. Your effort is really appreciated.
Comments 1: Please do not include abbreviations in the abstract, or mention what they mean: Results: The cohort included 44 GPA, 30 MPA, and 4 EGPA patients.
Response 1: We deleted the abbreviations in the abstract. Sorry for this mistake.
Comments 2: In summary, the following is unclear; please improve the following sentence: All deaths occurred in patients with pulmonary involvement (p = 0.048), and mortality tended to be higher in those with renal involvement (p = 0.065).
Response 2: We improved that sentence in the abstract.
Comments 3: In the methods section, define what was called pulmonary involvement (PI) or renal involvement (RI).
Response 3: We defined what was called pulmonary involvement (PI) or renal involvement (RI).
Comments 4: Please mention if there was committee approval, its approval number and date, as well as whether informed consent was required or waived, and whether the confidentiality of personal data was guaranteed.
Response 4: We added the approval date and number. Informed consent was waived, we added that information. Also we added the info about whether the confidentiality of personal data was guaranteed.
Comments 5: The description of the statistical analysis is very well done. Table 1. P values ​​of 1, 3, or 4 decimal places exist; please standardize them, also in the text. Please indicate which statistical test was used to obtain this P value.
Response 5: Thank you for this comment; we are very pleased to hear this. We have been working on the statistical analysis for a long time, and after such effort, it is truly rewarding to receive this feedback. We standardized the p values of Table 1, also in the text, and indicated the statistical test which we used to obtain this p value.
Comments 6: Post-hoc analyses are not shown; please mention them.
Response 6: We mentioned about post hoc analyses at the results part and highlighted them.
Comments 7: As this is a cohort study, please define the minimum, maximum, and average follow-up time of the patients. Also, in cohort studies, the RR is shown, and you mention OR; please verify.
Response 7: Thank you for this valuable comment. While this is a cohort study, the predictors of relapse and mortality were analysed using logistic regression models, which provide odds ratios (OR). Therefore, OR was used and reported accordingly. We will definitely use RR in our next studies and we hope we will meet again. We defined the minimum, maximum, and average follow-up time of the patients.
Comments 8: Please define everything you describe or analyze, for example, what you call elevated CRP levels, increased creatinine, hypoalbuminemia, etc.
Response 8: We defined everything we described or analyze and highlighted them in the methods section.
Comments 9: 3.4. Figures: It is not clear what you want to demonstrate, or the objective of this section; please expand.
Response 9: It was not clear what we want to demonstrate of this section, we expanded it.
Reviewer 3 Report
Comments and Suggestions for Authors
The manuscript, entitled „Pulmonary and Renal Predictors of Mortality in ANCA-Associated Vasculitis: A Regional Experience from Türkiye” by Dilara Bulut Gökten et al. describes a cohort of patients with ANCA associated vasculitis (GPA; MPA and EGPA) in a retrospective manner.
The evaluation focusses on the pulmonary and renal manifestation of the disease and relates this disease situation to the course of the disease, particularly regarding relapse and mortality. This is important information that has clinical significance, but the cohort is rather small compared to previously published cohorts. Nevertheless, I do recognize the importance of this work, particularly since other cohorts also come from other geographical areas. However, some aspects are not yet sufficiently well worked out, particularly regarding prognosis. In other areas, the work is too extensive and contains less important information.
Major points:
The new classification criteria should be used.
The two endpoints (relapse and death) should be presented in more detail
- when did they occur (also as a figure)?
- in which organs did the relapse occur?
- what caused the patients' deaths (activity or comorbidity)?
- how long was the follow-up period?
There are too many covariates in the regression analysis for a cohort of 78 patients
I recommend shortening the introduction and the discussion and the tables somewhat to make them more concise.
Could you please provide more information on organ biopsies and results.
Minor points:
- (page 3, line 104) IF is mentioned and ELISA (PR3 / MPO) is given in the results, please refer to ELISA tests
- For the MPO / PR3 double positive patients, did you check whether there was an association with medication or drugs?
- (table 1) is this only induction treatment, otherwise please indicate?
- (table2) 40% of MPA patients have pulmonary nodules? What kind of manifestation is this? If there are pulmonary granuloma, then please check if this is GPA.
Author Response
We sincerely thank you for your thoughtful and encouraging comments. We truly appreciate the recognition of our study’s importance despite the modest cohort size, and we value the constructive suggestions provided to help us improve the clarity and focus of our manuscript, particularly regarding prognostic aspects.
Comments 1: The new classification criteria should be used.
Response 1: We sincerely thank the reviewer for this valuable comment. We apologise for the oversight in our initial manuscript; we indeed classified our patients according to the newly published ACR/EULAR 2022 classification criteria, but we mistakenly referenced the older criteria in the text. We have now corrected this in the Methods section to accurately reflect the classification system used.
Comments 2: The two endpoints (relapse and death) should be presented in more detail
- when did they occur (also as a figure) ?- in which organs did the relapse occur?
- what caused the patients' deaths (activity or comorbidity)?- how long was the follow-up period?
Response 2: We presented the two endpoints in more detail according to your valuable comments.
Comments 3: There are too many covariates in the regression analysis for a cohort of 78 patients.
Response 3: We deleted some covariates in our analysis, deleted from Table 4, also added this limitation to our limitations paragraph. We will carefully consider limiting the number of covariates in future analyses and studies to enhance the robustness and generalisability of the findings.
Comments 4: I recommend shortening the introduction and the discussion and the tables somewhat to make them more concise.
Response 4: We shortened the introduction, the discussion and the tables.
Comments 5: Could you please provide more information on organ biopsies and results.
Response 5: We provided more information on organ biopsies and results.
Comments 6: IF is mentioned and ELISA (PR3 / MPO) is given in the results, please refer to ELISA tests
Response 6: We referred to ELISA tests.
Comments 7: For the MPO / PR3 double positive patients, did you check whether there was an association with medication or drugs?
Response 7: Thank you for this thoughtful question. We evaluated drug histories among MPO/PR3 double-positive patients but did not identify any associations with known ANCA-inducing medications such as hydralazine, propylthiouracil, or minocycline in our cohort.
Comments 8: (table 1) is this only induction treatment, otherwise please indicate?
Response 8: Table 1 includes both induction and maintenance immunosuppressive treatments administered during the disease course. We revised the table legend accordingly to explicitly reflect this information.
Comments 9: (table2) 40% of MPA patients have pulmonary nodules? What kind of manifestation is this? If there are pulmonary granuloma, then please check if this is GPA.
Response 9: Thank you for this valuable observation. Although pulmonary nodules are typically associated with granulomatous inflammation seen in GPA, we confirmed the MPA diagnoses based on the 2022 ACR/EULAR classification criteria, integrating clinical, serological, and histopathological data. No reclassification to GPA was necessary after review. We recognise this as a limitation and will consider a more in-depth re-evaluation of such cases in future studies.
Round 2
Reviewer 1 Report
Comments and Suggestions for Authors
Dear Authors,
Thank you for your thorough and thoughtful responses to the reviewer comments. I appreciate the effort you have made in revising the manuscript and addressing the points raised during the review process.
Reviewer 2 Report
Comments and Suggestions for Authors
The suggestions were attended to and the manuscript improved. I suggest publishing it
Reviewer 3 Report
Comments and Suggestions for Authors
The manuscript has become much clearer as a result of the revision. The information in the tables is still very extensive, partly redundant and partly of secondary importance for understanding the results. The authors have addressed the main points of my review. In this respect, the paper can now be published as it stands.